# BagelScore: Visual-Language Evaluation Made Easy

## Abstract

Evaluation remains a fundamental challenge in multimodal learning. Existing metrics such as CLIPScore, LPIPS, and FID reduce assessment to embedding similarity or perceptual distance, which systematically fails to capture semantic correctness or editing plausibility, while GPT-based scoring remains subjective and inconsistent. We argue that the emergence of bottleneck-free unified multimodal models enables a new evaluation paradigm: their internal reasoning and generative dynamics can serve as principled signals. Building on BAGEL, we propose two complementary metrics. BagelScore focuses on image understanding and image-text matching, outperforming traditional metrics like CLIPScore, LPIPS, FID, and GPT-based heuristics by directly evaluating the semantic alignment between images and captions using the unified model's reasoning capabilities. EditingScore, the first evaluation metric specifically designed for assessing image editing quality, quantifies the difficulty of learning the transformation in the latent space of a generative model. EditingScore is validated on Edit-1K, the first benchmark dataset specifically created for image editing quality evaluation. Together, BagelScore and EditingScore provide a unified, reasoning-based paradigm for multimodal evaluation.

## 1 Introduction

The rapid progress of foundation models has reshaped the multimodal landscape. Modern vision-language models (VLMs) exhibit impressive reasoning and generation capabilities, suggesting that evaluation should no longer be restricted to shallow similarity metrics. As models internalize rich world knowledge and semantic structure, their own representations and learning dynamics present a new opportunity: using foundation models themselves as reliable evaluators of multimodal quality.

Existing evaluation methods, however, remain limited. For image-text similarity, CLIPScore Hessel et al. (2022) computes cosine similarity between embeddings. While effective for literal captioning, this approach conflates geometric closeness with semantic correctness, failing on negations, substitutions, and compositional errors. For image editing, LPIPS and FID are widely adopted but measure only perceptual or distributional differences, leaving unanswered the central question of whether an edit is reasonable. GPT-based scoring has been proposed to incorporate semantics, but judgments are subjective, prompt-sensitive, and inconsistent across annotators. Together, these metrics fall short of robust, principled evaluation.

A recent architectural breakthrough opens a new path. Bottleneck-free unified multimodal models allow vision and language signals to interact within a single transformer, avoiding the projection bottlenecks of CLIP-style systems that compress away semantic detail. BAGEL exemplifies this class, integrating reasoning-oriented experts with flow-matching experts for generative modeling through a Mixture-of-Experts design. This architecture equips BAGEL not only to generate, but also to judge—preserving fine-grained semantics, supporting long-context reasoning, and enabling evaluation tasks that traditional models cannot reliably perform.

Building on this foundation, we introduce BagelScore, a reasoning-based metric for image-text similarity. Instead of relying on embedding similarity, BagelScore directly utilizes the BAGEL model's understanding capabilities to evaluate whether an image and a text describe the same content, deriving a compatibility score based on the model's reasoning. This approach transforms similarity

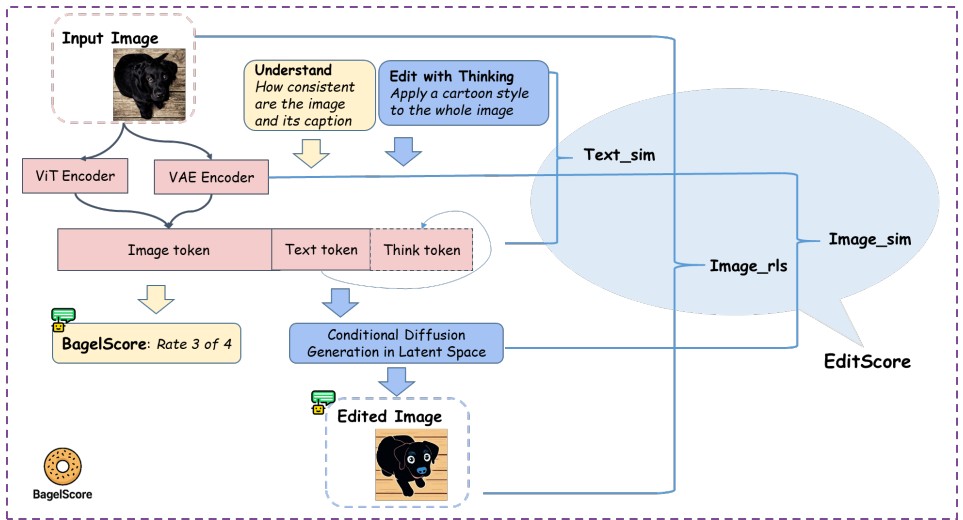

Figure 1: Overview of BagelScore and EditingScore Framework for Image-Text Alignment and Editing Evaluation

assessment into semantic judgment, capturing fine-grained errors that traditional embedding-based methods often miss.

For image editing, we propose EditingScore, the first principled metric designed to evaluate edit quality. EditingScore quantifies the difficulty of learning the transformation from a source image to an edited target in the latent space of a generative model. High-quality edits correspond to minimal latent space shifts, while poor edits require large, arbitrary transformations. This metric provides an objective measure of editing plausibility by operationalizing edit quality as the learnability of transformations, offering a scalable and principled alternative to perceptual or heuristic-based evaluations.

Together, BagelScore and EditingScore establish a unified, reasoning-based paradigm for multimodal evaluation. BagelScore improves image-text alignment beyond methods like CLIPScore, while EditingScore offers the first robust evaluation of image edits. Validated on standard benchmarks and our newly introduced Edit-1K dataset, EditingScore achieve good alignment with human judgment. More broadly, our findings highlight that as foundation models evolve, their internal reasoning and generative capabilities should serve as the foundation for future evaluation frameworks.

In this work, we make the following contributions:

- **BagelScore for semantic alignment.** We introduce BagelScore, a reasoning-based metric that reframes image-text similarity as a semantic judgment task. By leveraging BAGEL's bottleneck-free unified architecture and reasoning perplexity, BagelScore overcomes the limitations of CLIPScore and achieves stronger correlation with human judgment.

- **EditingScore for edit quality.** We introduce *EditingScore*, a principled metric for evaluating image edits by quantifying the difficulty of learning the transformation in the latent space of a generative model. EditingScore combines metrics like image cosine similarity, latent shift, and text similarity to assess the plausibility and consistency of edits.

- **Comprehensive validation.** We establish *Edit-1K*, a benchmark dataset for edit quality, and conduct extensive experiments showing that EditingScore achieve strong alignment with human judgment.

## 2 RELATED WORK

**Unified Multimodal Learning.** Unified multimodal learning has attracted considerable attention due to its goal of integrating diverse modalities, such as text, image, and video, into a single model. Early models like CLIP (Radford et al., 2021) established shared embedding spaces for text and

images, enabling zero-shot capabilities. Models like Flamingo (Alayrac et al., 2022) expanded this concept to few-shot learning, leveraging large language models for multimodal tasks. Recent advancements focus on unifying both understanding and generation within a single framework. These models can be broadly classified into three categories based on their backbone architectures: Diffusion (Shi et al., 2025; Yang et al., 2025; Swerdlow et al., 2025), MLLM (AR) (Wang et al., 2025; Lin et al., 2025; Wu et al., 2025c;b), and MLLM (AR + Diffusion) (Xie et al., 2025; Deng et al., 2025). Each of these categories is further subdivided according to the encoding strategy used, including Pixel Encoding (Wang et al., 2025; Xie et al., 2025; Zhou et al., 2024), Semantic Encoding (Wu et al., 2025a), Learnable Query Encoding (Xu et al., 2025), and Hybrid Encoding (Deng et al., 2025; Qu et al., 2025). BAGEL (Deng et al., 2025), a prominent open-source model, advances the field by enabling seamless multimodal understanding and generation through its Mixture-of-Transformer-Experts architecture. It achieves state-of-the-art performance across various tasks, including image editing and text generation, surpassing existing models on multiple benchmarks. Innovations such as Janus (Chen et al., 2025) have refined tokenization strategies and cross-modal attention mechanisms, addressing the challenges inherent in multimodal integration. Building on these developments, BagelScore aims to provide a comprehensive framework for evaluating multimodal tasks, offering a robust method for assessing the effects of image and text generation edits.

**Image Editing and Generation.** Recent progress in image editing and generation has been primarily propelled by diffusion-based frameworks and refined multimodal guidance, with two core paradigms—content-aware editing and content-free customization—emerging as the backbone of current research (Shuai et al., 2024; Huang et al., 2025). Content-aware editing focuses on modifying images while preserving or leveraging existing content, encompassing key tasks: object/attribute manipulation (addressed by models that integrate language and diffusion capabilities to follow editing instructions (Brooks et al., 2023) and strategies for precise attribute adjustment (Zhang et al., 2025)), inpainting (enabled by exemplar-guided methods with self-supervised training for semantically consistent region filling (Yang et al., 2022)), and cross-domain translation (facilitated by frameworks for efficient multimodal knowledge transfer with minimal parameters (Huang et al., 2021)). Methodologically, content-aware editing splits into training-free techniques—such as attention control for region-specific manipulation and score distillation for subtle edits (Hertz et al., 2022; Brack et al., 2024)—and training-based approaches that fine-tune models on task-specific data (Huang et al., 2023). In contrast, content-free customization targets personalized generation, covering subject-driven tasks (which retain object identity in novel generations (Ruiz et al., 2023)) and attribute-driven tasks (which control style attributes in generated content (Huang et al., 2024)); this paradigm relies on training-free tools for efficient low-rank adaptation (Hu et al., 2021). Collectively, these advancements reflect the field's shift toward controllable, user-centric solutions, and ongoing efforts to integrate advanced inversion and editing algorithms further address complex real-world scenarios—underscoring the value of our proposed BagelScore (a multimodal scoring method) for evaluating result quality and alignment in such contexts (Shuai et al., 2024; Gal et al., 2022; Sohn et al., 2023).

## 3 UNIFY MODEL EVALUATION SCORE

We address two distinct evaluation challenges in multimodal systems: (1) improving image-text similarity assessment beyond existing methods like CLIPScore through reasoning-based evaluation, and (2) introducing a novel approach for evaluating image editing quality where no principled methods currently exist. Both tasks leverage BAGEL, a Scalable Generative Cognitive Model with 7B active parameters trained on large-scale interleaved multimodal data, which enables complex reasoning over multimodal contexts through its Mixture-of-Transformer-Experts architecture with shared self-attention across modalities.

### 3.1 REASONING PARADIGM FOR IMAGE-TEXT CONSISTENCY

CLIPScore Hessel et al. (2022) evaluates image-text similarity by computing cosine similarity between visual and textual embeddings:

$$S_{\text{CLIP}}(x, y) = \cos(f_v(x), f_t(y)), \tag{1}$$

where $f_v(x)$ and $f_t(y)$ denote the vision and text encoder outputs, respectively. While effective on literal captioning tasks, this geometric matching approach conflates compositional similarity with semantic correctness. For instance, "a dog sitting" and "a cat sitting" may receive high scores due to similar visual arrangements, despite describing entirely different entities. As a result, CLIPScore systematically fails to detect semantic errors such as negations ("no cat" vs. "cat"), entity substitutions ("dog" vs. "cat"), and relational mistakes ("cat on table" vs. "table on cat"). Moreover, because CLIPScore lacks contextual and world knowledge, it performs poorly in domains such as news captions that demand richer reasoning. Fundamentally, CLIPScore answers the question *"how similar are the features?"* rather than *"is the semantics correct?"*.

**BagelScore: A Reasoning-based Metric for Multimodal Understanding.** The evaluation of image-text similarity remains a challenge in multimodal learning. Traditional metrics like CLIP-Score Hessel et al. (2022) and LPIPS Zhang et al. (2018) focus on embedding distances or perceptual differences but often fail to capture semantic correctness, especially for complex relationships between modalities. We introduce BagelScore, a novel metric that leverages the reasoning capabilities of the BAGEL model to assess the semantic alignment between images and captions.

The BagelScore is computed on a scale defined by the task, e.g., between 1 and 4 for the expert dataset. This approach overcomes the limitations of traditional metrics by offering a finer, more semantically grounded evaluation. The model's reasoning captures subtle mismatches such as negations, entity substitutions, and relational errors.

In summary, BagelScore provides a more reliable and principled evaluation of image-text alignment, leveraging the BAGEL model's understanding mode to align more closely with human judgment.

Below is an example of the prompt used for evaluation:

---

**Flickr8K-CF Prompt**

Analyze how accurately this image matches the caption: "caption"
Evaluate the match on a precise scale with only these four possible values:

- 0.0: No match - The caption is completely unrelated to the image.
- 0.3: Poor match - The caption has minimal relevance with major inaccuracies.
- 0.6: Good match - The caption is mostly accurate with only minor omissions or imprecisions.
- 1.0: Perfect match - The caption perfectly describes all important aspects of the image.

Consider:

- Factual accuracy (are all statements true about the image?)
- Completeness (does the caption cover the main elements?)
- Precision (does the caption avoid vague descriptions?)
- Relevance (does the caption focus on important aspects?)

Provide ONLY a single decimal rating (0, 0.3, 0.6, or 1.0) without explanation:

---

## 3.2 IMAGE EDITING QUALITY ASSESSMENT

Unlike image-text similarity, where metrics like CLIPScore are well-established, image editing quality lacks principled evaluation methods. Existing perceptual metrics such as LPIPS and FID focus on low-level similarities—LPIPS measures pixel-wise differences, and FID captures distributional divergence. While effective for detecting distortions, these metrics overlook the critical question: *is the edit reasonable?* Recent GPT-based scoring has attempted to assess semantic plausibility, but such evaluations are subjective, prompt-sensitive, and inconsistent. Current methods fail to capture high-quality editing criteria, including physical plausibility (adhering to real-world constraints), semantic consistency (preserving core object properties), and causal coherence (ensuring logical transformations). This underscores the need for evaluation approaches that consider the meaningfulness of edits, beyond perceptual similarity.

**EditingScore: Achieving Intent with Minimal Latent Space Changes.** We introduce *EditingScore*, a framework for evaluating image edits by quantifying the difficulty of learning transformations in the latent space of generative models. High-quality edits require minimal latent space changes, preserving the structure of the source image $x_s$ and aligning it with the target image $x_t$. Poor edits involve larger, arbitrary transformations, which deviate from the model's learned priors and are harder to generalize.

To quantify the transformation difficulty, we define three key metrics: image cosine similarity, image relative latent shift, and text similarity. The first, image cosine similarity, measures the alignment between the latent representations of the source image $x_s$ and the edited image $\hat{x}_t$. It is computed as the cosine of the angle between their latent vectors, with higher values indicating better alignment. The formula is:

$$\text{cosine\_sim}(x_s, \hat{x}_t) = \frac{\langle f(x_s), f(\hat{x}_t) \rangle}{\|f(x_s)\|\|f(\hat{x}_t)\|}$$

where $f(x)$ is the latent feature of image $x$ and $\hat{x}_t$ is the edited image.

The second metric, image relative latent shift (rls), quantifies the extent of transformation in latent space. It is defined as the Euclidean distance between the latent vectors of the source and target images, normalized by the latent space's dimensionality $d$:

$$\text{rls}(x_s, x_t) = \frac{\|z_s - z_t\|_2}{\sqrt{d}},$$

where $z_s$ and $z_t$ are the latent vectors of the source and target images, respectively.

The final metric, text similarity, measures how well the generated image aligns with the input prompt. It is computed as the cosine similarity between the text embedding $p$ and the model's latent representation $\hat{f}(x_t)$ of the target image:

$$\text{sim\_text}(p, \hat{f}(x_t)) = \frac{\langle p, \hat{f}(x_t) \rangle}{\|p\|\|\hat{f}(x_t)\|}$$

The final *EditingScore* is computed by combining these metrics:

$$\text{EditingScore} = \frac{\text{cosine\_sim}(x_s, \hat{x}_t) \cdot \text{sim\_text}(p, \hat{f}(x_t))}{\text{rls}(x_s, x_t)^2 + \varepsilon}$$

where $\varepsilon$ is a small constant (e.g., $10^{-10}$) to prevent division by zero. The score is normalized to the range $[0, 1]$, with lower values indicating high-quality edits that maintain structural and semantic alignment with the original, while higher values reflect more complex, less plausible edits.

## 4  DATASETS AND METRICS

### 4.1  DATASETS FOR IMAGE-TEXT SIMILARITY EVALUATION

We evaluate BagelScore on three established benchmarks for image-text similarity assessment:

**Flickr8K-Expert**. This dataset contains 8,000 images from Flickr8K with expert human annotations for caption quality. Each image is paired with 5 reference captions and multiple candidate captions scored by expert annotators on a 1-4 scale for semantic correctness. The dataset contains 40,000 image-caption pairs with expert ratings, making it ideal for evaluating fine-grained semantic alignment.

**Flickr8K-CF (Counterfactual)**. A challenging variant of Flickr8K specifically designed to test robustness against semantic errors. This dataset includes 8,000 image-caption pairs where captions contain deliberate semantic mistakes such as negations ("no cat" instead of "cat"), entity substitutions ("dog" instead of "cat"), and compositional errors ("cat on table" instead of "table on cat").

Human annotators rated these counterfactual pairs, providing ground truth for semantic correctness evaluation.

**Composite**. A large-scale benchmark combining multiple image-text datasets, including MSCOCO, Flickr30K, and Visual Genome annotations. This dataset contains 150,000 image-caption pairs with human judgments aggregated from multiple annotators. The diversity of image content and caption styles makes it suitable for evaluating generalization across different domains.

## 4.2 EDIT-1K DATASET

To evaluate EditingScore, we introduce **Edit-1K**, a novel benchmark for image editing quality assessment. The dataset construction follows a systematic pipeline:

**Data Collection**: We gathered 1,000 source images from diverse categories, including portraits, landscapes, objects, and scenes from MSCOCO and Open Images. For each source image, we generated 5-10 edited versions using state-of-the-art editing methods, including InstructPix2Pix, SINE, and Ledits++, resulting in 7,500 source-target image pairs.

**Annotation Process**: Three expert annotators evaluated each edit based on five criteria: (1) *Editing Accuracy* – how accurately the intended edit was applied, (2) *Visual Quality* – the aesthetic quality and clarity of the image after editing, (3) *Content Preservation* – whether the core content and objects in the image are preserved, (4) *Style Consistency* – whether the style of the edit is consistent with the original image, and (5) *Overall Effect* – a holistic assessment that combines all the aforementioned factors. Annotations were made on a 0-1 scale, with detailed guidelines provided to ensure consistency.

**Quality Control**: Inter-annotator agreement achieved Krippendorff's $\alpha = 0.78$ for overall quality scores. Disagreements were resolved through discussion, and 10% of samples were re-annotated by a fourth expert for validation.

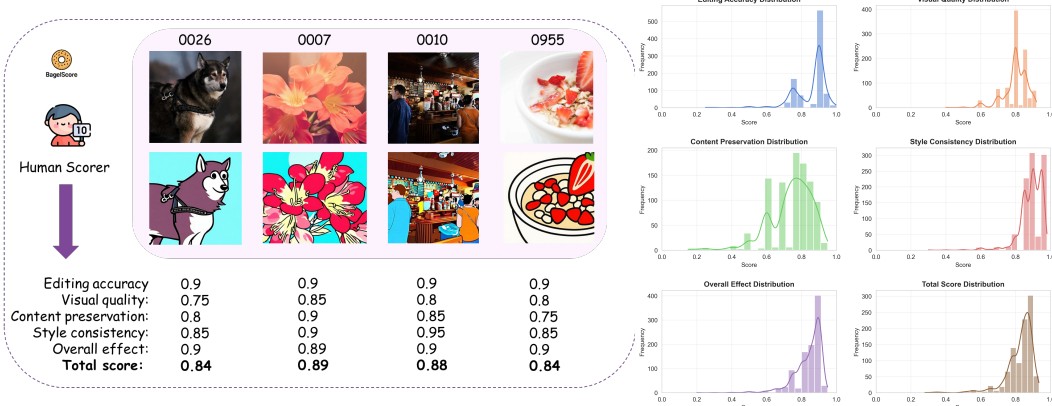

Figure 2: Edit-1K dataset and corresponding score distribution.

## 4.3 EVALUATION METRICS

**Kendall's Tau** ($\tau$). We use Kendall's tau correlation coefficient to measure ranking correlation between model scores and human judgments. This metric is robust to outliers and captures monotonic relationships, making it ideal for evaluating whether models can correctly rank image-text pairs or edits according to human preferences.

**Human Evaluation Studies**. Beyond automatic metrics, we conduct human evaluation studies where annotators compare model rankings with their own preferences, reporting agreement rates and preference scores to validate that our metrics align with human perception of quality.

## 5 EXPERIMENTS

### 5.1 IMPLEMENTATION DETAILS

All experiments were conducted on a system equipped with four NVIDIA A100 GPUs, each with 40GB of VRAM. The total system RAM is 256GB, providing ample memory for the large-scale processing required by the multimodal tasks. We used the PyTorch framework (version 1.10) for model implementation and training, utilizing the CUDA toolkit to leverage GPU acceleration. The models were trained using mixed-precision training with the `torch.cuda.amp` module, which helped optimize memory usage and computational efficiency.

### 5.2 MAIN RESULTS

Table 1 presents the correlation results of various image-text similarity metrics against human judgment, evaluated on the Flickr8K-Expert and Flickr8K-CF datasets. Notably, the performance of BagelScore stands out, achieving the highest correlation with human ratings across both datasets, surpassing traditional methods such as BLEU, METEOR, and CLIPScore. This demonstrates the advantages of leveraging modality merging through BagelScore, as opposed to the contrastive approach of CLIPScore.

In the Flickr8K-Expert dataset, which contains professionally annotated captions, BagelScore achieves a correlation of 53.2 ($\tau_c$), outperforming the best previous methods, RefCLIPScore (53.0) and ViLBERTScore-F (50.1). Similarly, for the more challenging Flickr8K-CF dataset, which includes counterfactual and semantically erroneous captions, BagelScore leads with a $\tau_b$ of 38.0, again surpassing RefCLIPScore (36.4) and CLIPScore (34.4). These results highlight that BagelScore, leveraging an integrated multimodal approach, effectively captures fine-grained semantic alignment, providing more reliable performance in both clean and error-prone settings.

| Metric | $\tau_c$ |
|---|---|
| BLEU-1 | 32.3 |
| BLEU-4 | 30.8 |
| ROUGE-L | 32.3 |
| BERTScore (RoBERTa-F) | 39.2 |
| METEOR | 41.8 |
| CIDEr | 43.9 |
| SPICE | 44.9 |
| ViLBERTScore-F | 50.1 |
| CLIPScore (no refs) | 51.2 |
| RefCLIPScore | 53.0 |
| **BAGELScore** | **53.2** |

(a) Flickr8K-Expert

| Metric | $\tau_b$ |
|---|---|
| BLEU-4 | 16.9 |
| CIDEr | 24.6 |
| METEOR | 22.2 |
| ROUGE-L | 19.9 |
| SPICE | 24.4 |
| BERTScore (RoBERTa-F) | 22.8 |
| LEIC* | 29.5 |
| CLIPScore (no refs) | 34.4 |
| RefCLIPScore | 36.4 |
| **BAGELScore** | **38.0** |

(b) Flickr8K-CF

| Metric | $\tau_c$ |
|---|---|
| BLEU-1 | 31.3 |
| BLEU-4 | 30.6 |
| ROUGE-L | 32.4 |
| BERT-S (RoBERTa-F) | 30.1 |
| METEOR | 38.9 |
| CIDEr | 37.7 |
| SPICE | 40.3 |
| BERT-S++ * | 44.9 |
| TIGEr | 45.4 |
| ViLBERTScore-F | 52.4 |
| CLIP-S (no refs) | 53.8 |
| RefCLIP-S | 55.4 |
| **BAGELScore** | **55.9** |

(c) Composite

Table 1: Correlations with human judgment on three datasets (Flickr8K-Expert, Flickr8K-CF and Composite). All metrics use 4–5 references, except CLIPScore which uses none. * indicates a result reported in prior work.

Table 2 presents the Pearson correlation coefficients between various evaluation metrics, including EditScore, Image RLS, Image Cosine Similarity, Text Similarity, Human Score, and GPT-based scoring for image editing. The Kendall Tau-b (**0.258872**) and Tau-c (**0.253286**) values reflect a moderate alignment between EditScore and human judgment, suggesting that EditScore captures relevant aspects of image edits, though there is still room for refinement. The higher correlation between EditScore and Image Cosine Similarity ($\tau_c = 0.78$) indicates that EditScore effectively captures the structural and semantic similarity between images.

Figures 3 (a) and (b) further demonstrate the consistency of EditScore with human ratings. Figure 3a shows a strong rank consistency between EditScore and Human Score, while Figure 3b illustrates

how EditScore aligns with Image Cosine Similarity, Text Similarity, and Image RLS in a 3D space. This visualization reinforces the idea that EditScore is most closely related to image structural similarity, supporting its effectiveness as an evaluation metric for image edits.

Table 2: Pearson Correlation Coefficients Between Evaluation Metrics

| Metric | EditScore | Image RLS | Image Cosine | Text Sim. | Human Score |
|---|---|---|---|---|---|
| EditScore | 1.00 | -0.78 | 0.78 | 0.05 | **0.14** |
| Image RLS | -0.78 | 1.00 | -0.74 | 0.00 | -0.12 |
| Image Cosine Sim. | 0.78 | -0.74 | 1.00 | 0.01 | 0.09 |
| Text Similarity | 0.05 | 0.00 | 0.01 | 1.00 | 0.05 |
| Human Score | **0.14** | -0.12 | 0.09 | 0.05 | 1.00 |

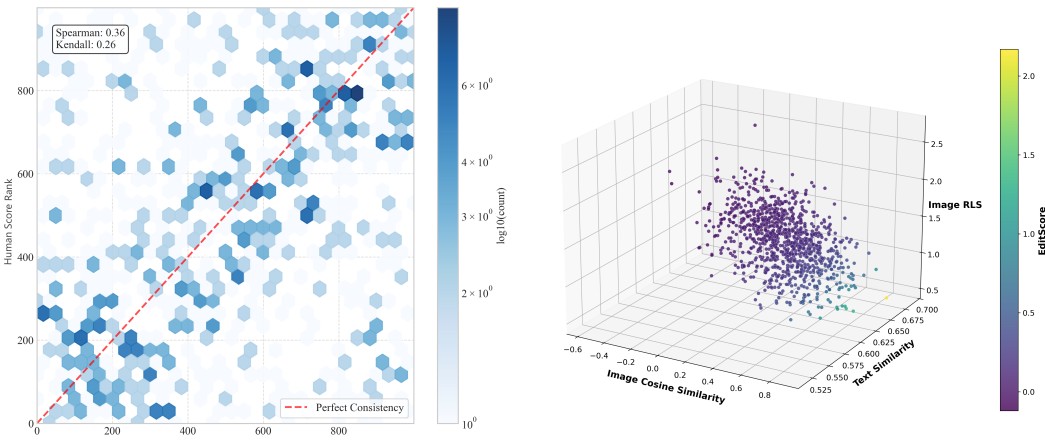

(a) Rank Consistency between Edit Score and Human Score.

(b) 3D Visualization of Edit Score with Image Cosine Similarity, Text Similarity, and Image RLS.

Figure 3: Comparison of Edit Score Consistency and 3D Visualization of Edit Score.

We compare EditScore with GPT-based scoring for image edits. As shown in Table 3, GPT-based scores generally show lower correlation with human judgment compared to EditScore. This highlights the advantage of EditScore in providing a more consistent and principled evaluation of image edits.

Table 3: Comparison of Kendall Tau Correlations between Human Judgment, EditScore, and GPT-based Scores

| Metric | Kendall Tau-b | Kendall Tau-c |
|---|---|---|
| Human Score | 1.000 | 1.000 |
| EditScore | **0.259** | **0.253** |
| GPT-based Score | 0.192 | 0.189 |

To assess the contribution of each component in *EditingScore*, we conducted an ablation study by removing one of the three key metrics: image cosine similarity, image relative latent shift, and text similarity. The results, shown in Table 4, reveal that removing either the image cosine similarity or image relative latent shift notably decreases alignment with human judgment, highlighting their importance in capturing structural and semantic consistency in edits. However, removing the text similarity metric results in only a minor decrease in performance, indicating that it plays a supplementary role in enhancing evaluation accuracy.

Table 4: Ablation Study on the Impact of Each Component in EditingScore

| Metric Removed | Kendall Tau-b | Kendall Tau-c |
| --- | --- | --- |
| None (Full EditingScore) | **0.259** | **0.253** |
| Image Cosine Similarity | 0.203 | 0.190 |
| Image RLS | 0.214 | 0.198 |
| Text Similarity | 0.238 | 0.229 |

## 6 DISCUSSION

Our study and results demonstrate several key implications for the future of multimodal learning.

**Advancing beyond shallow similarity metrics.** Traditional approaches, such as CLIPScore, LPIPS, and FID, have reduced multimodal evaluation to geometric or perceptual similarity, often missing semantic correctness or edit plausibility. BagelScore offers a fundamental shift by reframing image–text alignment as a semantic judgment task, leveraging the reasoning capabilities of unified multimodal models. Similarly, EditingScore provides the first principled metric for evaluating image edits by operationalizing the plausibility of transformations in latent space. These findings underscore the importance of using foundation models not only as generators but also as evaluators.

**Alignment with human judgment.**

Across multiple benchmarks, BagelScore consistently outperforms prior metrics, capturing fine-grained semantic errors such as negation and entity substitution. EditingScore, validated on the newly introduced Edit-1K dataset, achieves moderate yet meaningful correlation with human ratings, surpassing GPT-based heuristics in reliability. Notably, the ablation study (Table 4, p. 9) highlights the central role of structural similarity (cosine similarity and latent shift) in aligning with human perception, while text similarity contributes more marginally. This suggests that evaluation of edits relies more heavily on structural coherence than on textual alignment.

## 7 LIMITATION AND FUTURE WORK

A potential limitation of this work is the focus on static image datasets, which limits the scope of our evaluation metrics to image-text and image editing tasks. Extending our framework to include video and audio modalities is a natural next step, enabling a more comprehensive evaluation across multimodal content. Additionally, while our experiments utilize well-established datasets, exploring more diverse and challenging datasets will further assess the robustness of our metrics. Future work will also investigate the applicability of our evaluation metrics to other types of content transformation, such as text generation and multimodal synthesis, to enhance their versatility and broader applicability in multimodal AI tasks.

## 8 CONCLUSION

In this work, we introduce two novel evaluation metrics, BagelScore and EditingScore, as part of an effective attempt to leverage unified multimodal models for multimodal evaluation. BagelScore utilizes the reasoning capabilities of the unified BAGEL model to assess image-text similarity, outperforming traditional metrics such as CLIPScore by focusing on semantic alignment rather than embedding similarity. EditingScore, the first principled metric for evaluating image editing quality, quantifies the difficulty of learning transformations in the latent space of generative models, offering a more objective and scalable approach to evaluate editing plausibility. BagelScore is validated on established benchmarks, while EditingScore is validated on the newly introduced Edit-1K dataset, specifically designed to evaluate image editing quality. Both metrics demonstrate strong alignment with human judgment, establishing a unified, reasoning-based framework. These contributions underscore the promise of utilizing foundation models' internal reasoning to enhance the reliability and sophistication of evaluation methods across a wide range of multimodal tasks.

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
