# OpenReview forum: "BagelScore: Visual-Language Evaluation Made Easy"
_ICLR.cc/2026/Conference — ICLR 2026 Conference Withdrawn Submission_

### Official Review · Reviewer_mYua · 2025-10-24

**Soundness:** 2
**Presentation:** 2
**Contribution:** 2
**Rating:** 2
**Confidence:** 3

**Summary:**

This paper identifies challenges in existing multi-modal (image-text) evaluation methods, and proposes two new metrics for vision-language evaluation: BagelScore, and EditingScore. Results on existing benchmarks and the new Edit-1K benchmark validate the effectiveness of these scores relative to existing evaluation methods and human evaluation.

**Strengths:**

Vision-language evaluation is an important topic in general, and the authors correctly acknowledge some existing gaps in the ability to effectively perform such evaluation.

Both BagelScore and EditingScore do appear to be somewhat effective at the given task, relative to the methods compared to. EditingScore proposes a potentially interesting combination of scores which appears novel, though lacking theoretical justification.

Edit-1K is a worthwhile contribution as new manually-annotated data is generally useful to the community.

**Weaknesses:**

There are significant issues in this paper which I believe necessitate major revision.

This work positions itself as the “first” to present evaluation metrics and datasets for image editing (L22, 24, 77, 86, etc.), disregarding a significant body of recent work and existing benchmarks on this topic [1–4]. Similarly, BagelScore is very similar to existing methods which prompt VLMs to assess image-text alignment [5–9]. These are not discussed in the Related Work section or compared to.

There are several core methodological issues:
* The central model BAGEL is not clearly defined or cited, besides the vague descriptions around L44 and  L152. The use of this model is not justified or ablated, and it is unclear if its use is a central part of the method or if this could be used with other VLMs (which are not tested). The paragraph on L44 claims that the architecture of BAGEL is a key element enabling the system, but this is not clearly demonstrated.
* The paper claims EditingScore is “principled” (L77 etc.) but no theoretical justification is given for its ad-hoc formula (L250).
* Edit-1K, which is relatively small-scale with only 1K items and three annotators, contains human ratings on many axes, but evaluation does not specify which of these is compared with model outputs. There also lacks information about how the experts were sourced, if they were volunteers or compensated.
* Sec 5.2 presents “image-text similarity metrics” (L337) but listed metrics such as BLEU and ROUGE are text-text similarity metrics. It is unclear how these can be correlated with human ratings of text-image similarity.
* “GPT-based scoring” is compared to (e.g. Table 3) but never clearly defined.
* The overall performance of EditScore (τ=0.26, Table 3) seems rather low, raising questions about its effectiveness and/or the quality of the human annotations.
* There lack confidence intervals or significance tests. For example, it is not clear if the text similarity in EditScore has a statistically significant effect on performance from Table 4, given the limited sample size of Edit-1K.


There are also moderate presentation issues. For example, the Related Work section contains long lists of multimodal models and image editing methods, which are not directly relevant to the paper’s core topic (vision-language evaluation), while there is little coverage of vision-language evaluation methods. “EditingScore” and “EditScore” are used inconsistently, and Appendix A.2 is missing.

[1] Wang et al. Imagen Editor and EditBench: Advancing and Evaluating Text-Guided Image Inpainting. CVPR 2023

[2] Zhang et al. MagicBrush: A Manually Annotated Dataset for Instruction-Guided Image Editing. NeurIPS 2023

[3] Basu et al. EditVal: Benchmarking Diffusion Based Text-Guided Image Editing Methods. 2023

[4] Yosef et al. EditInspector: A Benchmark for Evaluation of Text-Guided Image Edits. ACL 2025

[5] Lu et al. LLMScore: unveiling the power of large language models in text-to-image synthesis evaluation. NeurIPS 2023

[6] Singh and Zheng. Divide, evaluate, and refine: Evaluating and improving text-to-image alignment with iterative vqa feedback. NeurIPS 2023

[7] Ku et al. VIEScore: Towards Explainable Metrics for Conditional Image Synthesis Evaluation. ACL 2024

[8] Lin et al. Evaluating Text-to-Visual Generation with Image-to-Text Generation. ECCV 2024

[9] Saxon et al. Who Evaluates the Evaluations? Objectively Scoring Text-to-Image Prompt Coherence Metrics with T2IScoreScore (TS2). NeurIPS 2024

**Questions:**

In Figure 1, which elements are part of BagelScore or Edit(ing)Score? Is EditScore really part of BagelScore as indicated by the dotted box? If is there an arrow from the “text token” to a Conditional Diffusion model, does this mean that it is conditioned on the same model (BAGEL) that is used for BagelScore? What do the small robot emojis mean?

Why does this work use BAGEL rather than other VLMs? Is the paper claiming that BAGEL would perform better (which is not justified or tested), or if this would work for other VLMs, why are they not tested?

Why do Tables 1-3 present different methods with Pearson vs. Kendall correlation, rather than reporting both metrics for all methods? Similarly, why are different methods shown in Table 1 a-c rather than showing the results of all methods on all datasets?

---

### Official Review · Reviewer_oCvu · 2025-10-28

**Soundness:** 2
**Presentation:** 2
**Contribution:** 2
**Rating:** 2
**Confidence:** 4

**Summary:**

To evaluate image-text pair similarities, this paper presents two different scores, BagelScore and EditingScore. The former one leverages the Bagel model to reason, while the latter one considers the editing ability of image-text pairs. The authors further introduce a novel dataset, Editing-1K.

**Strengths:**

1. The paper is easy to follow.
2. The experimental results are somewhat compelling.

**Weaknesses:**

1. The experimental section lacks some important compared methods, like PAC-S [1] and HICE-S [2].
2. The results of BagelScore in Table 1 surpass the reference-based RefCLIPScore, but it still lags reference-free PAC-S and HICE-S.
3. The reference forms in the paper should be checked again and again.
4. Prompts can be listed in the appendix.

[1] Positive-Augmented Contrastive Learning for Image and Video Captioning Evaluation, CVPR 2023.

[2] HICEScore: A Hierarchical Metric for Image Captioning Evaluation, ACM MM 2024.

**Questions:**

1. From the Introduction part, I cannot fully understand the motivation of EditingScore, especially Line 79 – Line 83.
2. More motivations and details about the Edit-1K dataset should be included in the Introduction part, not just a direct mention.
3. For complex relationships between modalities mentioned in Line 174, both HICE-S and InfoMetIC [3] proposed several solutions.
4. Can we use EditingScore to evaluate datasets in Table 1?
5. Which models do we need to obtain EditingScore? Is it just the BAGEL model? Or do we need more models?

[3] InfoMetIC: An Informative Metric for Reference-free Image Caption Evaluation, ACL 2023.

---

### Official Review · Reviewer_XmPf · 2025-10-28

**Soundness:** 2
**Presentation:** 2
**Contribution:** 2
**Rating:** 2
**Confidence:** 3

**Summary:**

The paper introduces two evaluation scores: BagelScore for image-text similarity and EditingScore for image editing. Both are based on the BAGEL model, a unified multimodal framework designed for evaluating visual-language tasks. They also introduce a 1K-Edit dataset which is a dataset for editing task.

**Strengths:**

The paper tackles an important problem in multimodal research, focusing on how to more effectively evaluate image-text similarity and image editing models. It introduces new metrics and a dataset that can serve as a strong foundation for future studies in this area. The main idea is well motivated. Rather than depending solely on feature-based similarity measures such as CLIPScore, the authors employ a unified vision-language model that jointly interprets images and text to assess semantic alignment more directly. The proposed EditingScore integrates three intuitive aspects: visual similarity between images, the degree of change in the model’s internal representation, and consistency with the target text prompt. The inclusion of ablation studies further clarifies the contribution of each component. Overall, the paper identifies important gaps in evaluation of multimodal AI and takes promising steps toward more semantically meaningful and reliable assessment methods.

**Weaknesses:**

The authors criticize GPT-based scoring as “subjective, prompt-sensitive, and inconsistent,” yet their own approach relies on exactly the same principle by prompting BAGEL to rate images on a scale. The only difference is a carefully engineered prompt, not a genuine methodological advancement. There is no concrete algorithmic or theoretical contribution; BagelScore is simply another instance of using an LLM as a judge, repackaged with different wording.
The quantitative results are equally weak. In Table 1, the reported performance margins are trivial and likely fall within normal measurement error. No significance testing is provided, yet the authors claim to “outperform” prior metrics. In Table 3, the correlation with human judgment is extremely poor (0.26) despite being described as “strong alignment.” Both EditingScore and the GPT baseline fail to meaningfully track human perception.
The mathematical logic is also inconsistent. The paper defines lower EditingScore values as indicating higher quality, but Table 2 shows a positive correlation (0.14) with human scores, implying the opposite relationship.
Furthermore, the paper removes essential comparisons with perceptual and editing metrics that would make the evaluation meaningful. Although it criticizes metrics such as LPIPS and FID, it fails to include them as baselines. For EditingScore, it should also be compared against established alternatives such as SSIM (Structural Similarity), DINO (https://openaccess.thecvf.com/content/ICCV2021/papers/Caron_Emerging_Properties_in_Self-Supervised_Vision_Transformers_ICCV_2021_paper.pdf ), and PickScore (https://arxiv.org/pdf/2305.01569), which are used in recent image editing evaluation. There are also more recent evaluation setups that could have been used instead of replicating the old CLIPScore design. For example, EditBench and PIE-Bench provide modern frameworks for assessing both edit quality and faithfulness.
Finally, there is no robustness analysis exploring how sensitive BagelScore is to prompt variations or different LLMs such as GPT-4V, Claude, or Gemini. Without such experiments, the claimed reliability and generality of the metric remain unsupported.

**Questions:**

1-Can you provide comparison results for the following metrics on your datasets: DINO similarity, SSIM, and LPIPS/FID?
2- Could you test your prompt with other vision-language models besides BAGEL and compare the results?
3-Figure 1 shows a schematic view, but there is no explanation in the text. Could you clarify what it represents and how it connects to your method?
4-Could you explain the reasoning part behind your metric?

---

### Note · Authors · 2025-11-12

I have read and agree with the venue's withdrawal policy on behalf of myself and my co-authors.